# Towards the Identifiability and Explainability for Personalized Learner Modeling: An Inductive Paradigm

## ABSTRACT

Personalized learner modeling using cognitive diagnosis (CD), which aims to model learners' cognitive states by diagnosing learner traits from behavioral data, is a fundamental yet significant task in many web learning services. Existing cognitive diagnosis models (CDMs) follow the *proficiency-response* paradigm that views learner traits and question parameters as trainable embeddings and learns them through learner performance prediction. However, we notice that this paradigm leads to the inevitable non-identifiability and explainability overfitting problem, which is harmful to the quantification of learners' cognitive states and the quality of web learning services. To address these problems, we propose an identifiable cognitive diagnosis framework (ID-CDF) based on a novel *response-proficiency-response* paradigm inspired by encoder-decoder models. Specifically, we first devise the diagnostic module of ID-CDF, which leverages inductive learning to eliminate randomness in optimization to guarantee identifiability and captures the monotonicity between overall response data distribution and cognitive states to prevent explainability overfitting. Next, we propose a flexible predictive module for ID-CDF to ensure diagnosis preciseness. We further present an implementation of ID-CDF, i.e., ID-CDM, to illustrate its usability. Extensive experiments on four real-world datasets with different characteristics demonstrate that ID-CDF can effectively address the problems without loss of diagnosis preciseness. Our code is available at https://anonymous.4open.science/r/ID-CDF-AB86/.

## 1 INTRODUCTION

Recent years have witnessed the rapid emergence of various online learning platforms on the web, such as Coursera[1] and ASSIST-ments[2]. On these platforms, web users from different areas (e.g., lawyers, engineers, college students) can act as learners and enjoy various personalized learning services such as learning resource recommendation [40] and adaptive learning [42]. In these services, a fundamental yet significant component is personalized learner modeling [25, 32], which aims to model learners' cognitive states from their online behavioral data. A widely applied personalized learner modeling technique is the cognitive diagnosis (CD) [19], which utilizes cognitive diagnosis models (CDMs) to diagnose learners' traits that represent their cognitive states from response data (e.g., question scores). Then CDMs can provide diagnostic results to the platforms for personalized web learning services and return them to learners as the feedback of their learning performance.

As shown in Figure 1, CD-based personalized learner modeling follows a *proficiency-response* (**P-R**) paradigm, where CDMs are *score prediction-based* that transductively diagnose learners' latent traits and question parameters through predicting learner performance. In the first step, CDMs randomly initialize trainable learner

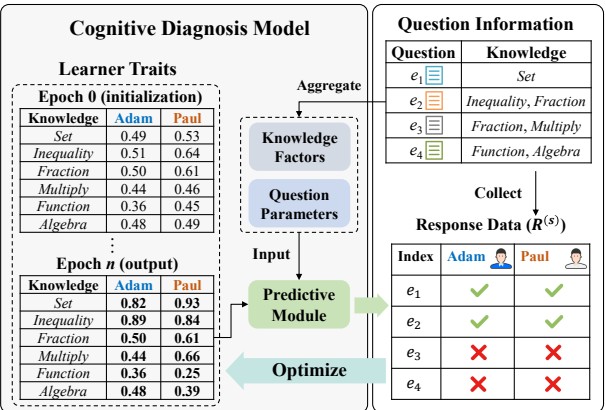

**Figure 1: An example of existing CD-based learner modeling. Learner traits ($\Theta$) and question parameters ($\Psi$) are fitted on the response data through transductive learning.**

traits and question parameters. Next, learner traits, question parameters, and pre-given knowledge factors are input into the predictive module of CDMs to predict response data. Finally, CDMs transductively diagnose learner traits and question parameters by parameter optimization. The *proficiency-response* paradigm is simple and easy for implementation in real-world scenarios, and has become the cornerstone of many classical CDMs like DINA [7] and NCDM [31].

Through our investigation, we find that based on the *proficiency-response* paradigm, CDMs suffer from the **non-identifiability** problem and **explainability overfitting** problem, which is harmful to the quantification of learners' cognitive states and limits their usability in web learning services. *Identifiability* plays a significant role in CD-based learner modeling and has been discussed in many works [36, 38], which connotes that diagnostic results should be able to distinguish between learners with different response data distribution. That is, different learner traits should lead to different response data distributions. Conversely, if a CDM cannot generate identifiable diagnostic results (i.e., learners with the same response data distribution have different diagnostic results), then it confronts the non-identifiability problem [37]. For example, in Figure 1, although diagnostic results from the CDM show that Adam and Paul have different learner traits, their response data distributions are the same. We notice that non-identifiability originates from the inevitable *randomness* in the parameter optimization of the *proficiency-response* paradigm. Specifically, in optimization, the random initialization and random update in the optimization algorithm (e.g., random sample order in mini-batch gradient descent) can cause different diagnostic results that generate the same prediction for learners with the same response data. The non-identifiability problem widely exists in existing CD-based learner modeling techniques. Figure 2 shows the histogram of *the Manhattan distance between diagnostic results of learners with the same response data* in a real-world dataset (see Appendix A.2 for description), diagnosed by NCDM [31]. We

---

[1]https://www.coursera.org/
[2]https://new.assistments.org/

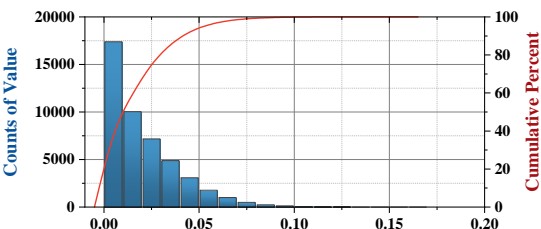

Figure 2: The histogram of the Manhattan distance of diagnostic results of learners with the same response distribution in Math1 dataset, diagnosed by NCDM [31].

can observe from the cumulative curve that over 50% of the distance is positive, which means that the diagnostic results of these learners are *unidentifiable*. With this problem, diagnostic results cannot be used to distinguish between learners with different learning performances, which affects the quality of web learning services from various aspects such as fairness in recommendation [4, 11].

Second, explainability is the ability that diagnostic results truly reflect *actual* cognitive states such as knowledge mastery levels, which plays an indispensable role in CD-based learner modeling [7, 27]. For example, in Figure 1, since Adam and Paul correctly answered all questions that require the mastery of *Set* and *Inequality*, the value of their diagnostic results on these knowledge concepts should be relatively high to reflect their proficiencies. Stemming from educational psychology, the explainability is ensured by the *monotonicity assumption* [27] between response scores and corresponding dimensions of learner traits because actual cognitive states are *latent* and *unobservable*. Along this line, researchers have devoted massive efforts to empower the explainability of CDMs. Some utilize elaborately designed monotonic interaction function [7], while others utilize parameter constraints [20, 31]. However, we notice in experiments for the first time that existing CDMs suffer from the *explainability overfitting problem*. That is, diagnostic results are highly explainable in observable response data for training, while less explainable in unobservable response data for testing. We discover that the problem originates from the *transductive learning process* in the *proficiency-response* paradigm. Specifically, CDMs only learn the monotonicity between diagnostic results and *observed* responses as the former are only optimized to predict *observed* responses in the training data. Therefore, they cannot capture the monotonicity between *overall* response data distribution and cognitive states. However, in web learning services, diagnostic results should represent the overall cognitive states of learners, which are reflected not only in training data but in other remaining response data. As a result, this problem limits the usability of diagnostic results in web learning services and needs to be solved.

To address these problems, we propose an *identifiable cognitive diagnosis framework* (**ID-CDF**) based on a novel *response-proficiency-response* (**R-P-R**) paradigm (see Figure 3) inspired by encoder-decoder models. Specifically, we first devise a diagnostic module to encode response data to learner traits and question parameters separately. This module utilizes our proposed identifiability condition and the monotonicity condition to simultaneously eliminate randomness in the optimization of diagnostic results and inductively capture the general monotonicity from data. Next, we utilize a flexible predictive module to reconstruct response data to

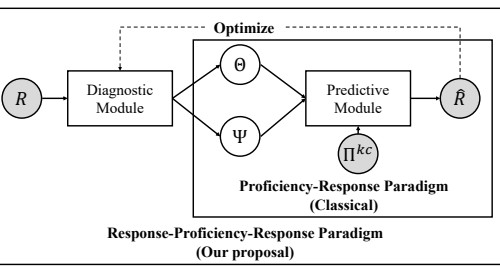

Figure 3: The classical P-R paradigm and our proposed R-P-R paradigm in personalized learner modeling using CD.

ensure the preciseness of diagnosis. Then, we present an implementation of ID-CDF, i.e., ID-CDM, to illustrate its usability. We demonstrate the effectiveness of ID-CDF in terms of identifiability, explainability, and preciseness by experiments on four public real-world learner modeling datasets with different characteristics. To the best of our knowledge, ID-CDF is the first framework that addresses the non-identifiability and explainability overfitting problem in the CD-based learner modeling task through innovation at the paradigm level, which can benefit various downstream web learning services with its high-quality personalized learner modeling capability.

The contribution of this paper is summarized as follows:
- We discover the non-identifiability and explainability overfitting problem in the existing CD-based learner modeling paradigm and illustrate for the first time the cause of these problems.
- We propose ID-CDF, which utilizes a novel *response-proficiency-response* paradigm to address the non-identifiability and explainability overfitting problem of CD-based learner modeling.
- We demonstrate the effectiveness of ID-CDF by applying its implementation ID-CDM to extensive experiments on four real-world datasets with different characteristics.

## 2 RELATED WORK

**Cognitive Diagnosis Model**. Existing CDMs are based on the *proficiency-response* paradigm. For instance, Deterministic Input, Noisy 'And' gate model (DINA) [7] is a discrete CDM that assumes knowledge mastery levels are binary, and utilizes a logistic-like interaction function to predict response scores from learner traits and question parameters. Item Response Theory (IRT) [2, 9] is a continuous CDM. In the two-parameter IRT (2PL-IRT) [9], a learner $i$'s ability is modeled as a scalar $\theta_i$, while a question $j$ is represented by its discrimination $a_j$ and difficulty $b_j$. Then, the response score given the learner's ability and the question parameter is modeled as $P(r_{ij} = 1|\theta_i, a_j, b_j) = \frac{1}{1+\exp\{-a_j(\theta_i-b_j)\}}$, where $r_{ij}$ denotes the response score. Learner abilities and question parameters are estimated by parameter optimization methods, such as full Bayesian statistical inference with MCMC sampling [12, 15] or variational inference [34]. Multidimensional Item Response Theory (MIRT) [27] further extends learner abilities and question difficulties to multidimensional cases, while the interaction function is still logistic-like. So far, deep learning techniques [31, 39] have also been widely applied to CD to reach a more accurate diagnosis. For instance, NCDM [31] leverages a three-layer positive full-connection neural network to capture the complex interaction between learners and questions. However, these CDMs would inevitably face the non-identifiability

problem and the explainability overfitting problem because of the limitation of the *proficiency-response* paradigm.

**Encoder-decoder Framework**. Stemming from statistical machine translation [5], the basic idea of the encoder-decoder framework is that a sequence of symbols (e.g., a sentence written in natural languages) can be encoded to a fixed-length vector that contains its semantics, and the vector can be decoded to the target sequence of symbols (e.g., translation) using its semantics. Because of its usability in semantic extraction and data reconstruction, the framework has been widely applied to many other fields like recommender systems [17, 21, 28, 35] and fake news detection [24, 26, 33]. For instance, in recommender systems, AutoRec [28] is an encoder-decoder collaborative filtering model that can encode latent user interests from observed rating score data and utilize the encoder output to predict (decode) unobserved ratings. Based on AutoRec, CDAE [35] introduces dropout [29] and user embedding at the input layer to get a better prediction of the rating matrix. CVAE [21] uses a Bayesian generative model to consider both rating and content (e.g., text) for recommendation in multimedia scenarios. However, despite their effectiveness in many fields, existing encoder-decoder models are not suitable for CD-based personalized learner modeling. First of all, existing encoder-decoder models only focus on the semantic extraction capability of the encoder while ignoring the explainability of encoder outputs, which is crucial in CD-based learner modeling and its downstream web learning services. Second, existing methods cannot model the complex interaction between learners and questions, which reflects learners' cognitive process and is indispensable in this scenario.

## 3 METHODOLOGY

### 3.1 Preliminary

In this part, we first present necessary mathematical notations. Then we define the CD-based learner modeling task. Finally, we define identifiability and monotonicity assumptions.

*3.1.1 Mathematical Notations and Task Definition.* To begin with, $S = \{s_1, s_2, \ldots, s_N\}$ denotes the learner set, where $N$ is the number of learners. $E = \{e_1, \ldots, e_M\}$ denotes the question set, where $M$ is the number of questions. $C = \{c_1, \ldots, c_K\}$ denotes the knowledge concept set, where $K$ is the number of knowledge concepts. $Q = (q_{jk})_{M \times K}$ denotes the question-knowledge mapping matrix manually labeled by experts, namely Q-matrix [30], which denotes what knowledge concepts are required by questions to correctly respond. For each component in the Q-matrix, $q_{jk} = 1$ denotes that question $e_j$ requires knowledge concept $c_k$ to correctly respond, otherwise $q_{jk} = 0$. For each item in response data, $r_{ij} \in \{0, 1\}$ denotes the dichotomous response score of learner $s_i$ on question $e_j$, where $r_{ij} = 1$ means a correct response while $r_{ij} = 0$ means an incorrect response. The total response data is a set of tuples, i.e., $\mathcal{D} = \{(s_i, e_j, r_{ij})|s_i \in S, e_j \in E, r_{ij} \in \{0, 1\}\}$. Learner traits, i.e., learners' cognitive states, are represented by $\Theta = \{\theta_i|s_i \in S\}$. Question features are represented by $\Psi = \{\psi_j|e_j \in E\}$. Next, the CD-based learner modeling task is defined as follows:

*Definition 3.1.* **CD-based Learner Modeling Task**. Given the response data set $\mathcal{D}$ and the Q-matrix $Q$, the goal of the task is to mine learner traits $\Theta$ as learners' cognitive states by modeling learners' performance prediction process.

*3.1.2 Identifiability and Explainability of CDMs.* In the literature of statistics [3], the identifiability of a distribution function implies that a set of parameters $\beta \in B$ of the distribution function set $\{f(\cdot|\beta)|\beta \in B\}$ is identifiable if distinct values of $\beta$ lead to distinct distribution, i.e., there does not exist other parameter value $\tilde{\beta} \neq \beta$ such that $f(\cdot|\tilde{\beta}) = f(\cdot|\beta)$. Similarly, in CD-based learner modeling, the identifiability of diagnostic results implies that distinct diagnostic results lead to distinct response distribution [36, 38]. Formally, the identifiability of diagnostic results is defined as follows:

*Definition 3.2.* **Identifiability in CD-based learner modeling**. Let $B = \{\Theta, \Psi\}$ be the set of diagnostic results, and let $\{f_R(\theta; \psi) : \Theta \times \Psi \to \{0, 1\}|\theta \in \Theta, \psi \in \Psi\}$ be the set of response function which generate response data given diagnostic results. Furthermore, let $r_i^{(s)} = f_R(\theta_i; \cdot)$ be the response data distribution of learner $s_i$ with trait $\theta_i$. Let $r_k^{(e)} = f_R(\cdot; \psi_k)$ be the response data distribution of question $e_k$ with feature $\psi_k$. Then the set of diagnostic results is identifiable if and only if distinct diagnostic results lead to distinct distribution of response data. Specifically, the identifiability of learner traits connotes that

$$\theta_i \neq \theta_j \to r_i^{(s)} \neq r_j^{(s)}. \tag{1}$$

In addition, the identifiability of question parameters connotes that

$$\psi_k \neq \psi_l \to r_k^{(e)} \neq r_l^{(e)}. \tag{2}$$

Then a set of diagnostic results is identifiable if both Eq.(1) and Eq.(2) are satisfied.

For instance, given the interaction function of a CDM, if the diagnostic results of two learners are different, then the distribution of their response data should also be different. Essentially, identifiability requires CDMs to be able to identify learners' response data from their diagnostic results.

*Definition 3.3.* **Explainability in CD-based learner modeling**. The explainability of learners' diagnostic results is defined as the ability they correctly reflect learners' actual cognitive states.

For example, if a learner has mastered the knowledge concept 'Inequality', then the component value of the diagnosed learner trait on this knowledge concept should be high so that the diagnostic result can correctly reflect the fact that the learner has mastered the knowledge concept. However, it is difficult to directly keep the explainability of diagnostic results because learners' true mastery levels are unobservable. As a result, in CD-based learner modeling, the explainability of diagnostic results is usually indirectly satisfied by the monotonicity assumption [27, 31]:

*Definition 3.4.* **Monotonicity assumption**. The probability of every learner correctly answering a question is monotonically increasing at any relevant component of his/her knowledge mastery level. Formally, the monotonicity assumption is equivalent to:

$$\theta_i^{(l)} \geq \theta_j^{(l)} \Leftrightarrow r_{il} \geq r_{jl}, \forall s_i, s_j \in S, \ e_l \in E, \tag{3}$$

where $\theta_i^{(l)}(s_i \in S, e_l \in E)$ denotes the relevant component of $s_i$'s knowledge mastery level $\theta_i$ to question $e_l$.

For score prediction-based CDMs, the monotonicity assumption usually depends on the monotonicity property of the interaction function [31]. For traditional CDMs such as DINA [7] and IRT [2], the interaction function is usually linear, thus inherently satisfying the monotonicity assumption. For deep learning-based CDMs such

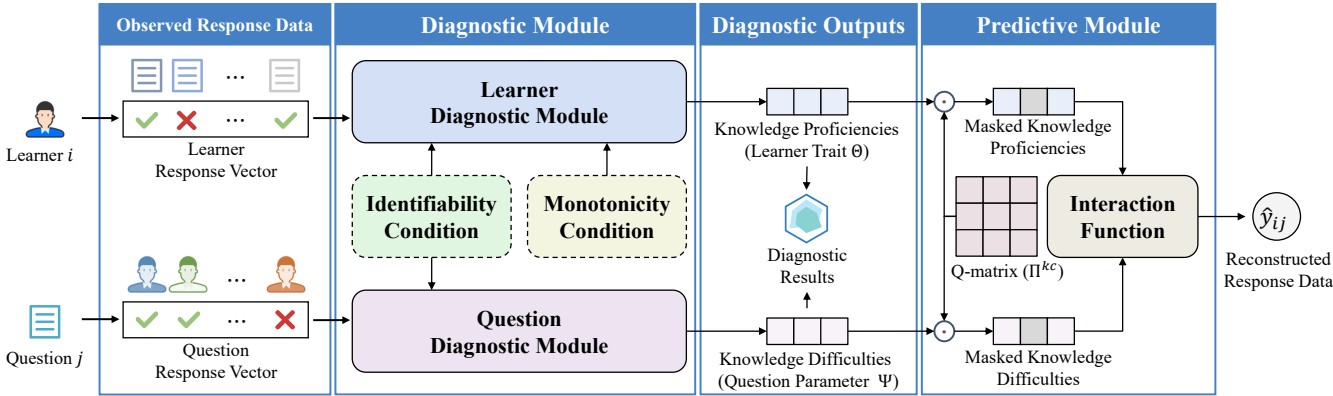

**Figure 4: The structure of identifiable cognitive diagnosis framework (ID-CDF).**

as NCDM [31], the weight parameter of the interaction function is limited to be non-negative to satisfy the assumption.

## 3.2 The Structure of ID-CDF

The structure of ID-CDF is shown in Figure 4. It is based on the *response-proficiency-response* paradigm, consisting of a diagnostic module and a predictive module. To address the non-identifiability and explainability overfitting problem, ID-CDF utilizes the diagnostic module with the identifiability and the monotonicity conditions to obtain an inductive estimation of diagnostic results. For the non-identifiability problem, we propose the identifiability condition, which limits the mapping relationship between response data and diagnostic results to satisfy the identifiability using its *contrapositive*. For the explainability overfitting problem, we propose the monotonicity condition to enable the diagnostic module to directly capture the monotonicity between overall response score distributions and learner traits. Then, ID-CDF utilizes the predictive module to model the complex interaction between learners and questions to ensure diagnosis preciseness.

**Diagnostic Module**. The diagnostic module aims to address the non-identifiability and explainability problem and inductively obtain diagnostic results from response data. To this end, the module utilizes diagnostic functions that satisfy the identifiability condition and the monotonicity condition to induce diagnostic results from vectorized response data.

Without loss of generality, we assume response data consists of response logs, which is common in web learning platforms [40]. In the first step, we transform response logs into response vectors that contain learners' latent cognitive states or questions' latent parameters (e.g., difficulty and discrimination). For learner $i$, let $\boldsymbol{x}_i^{(s)} = (x_{i1}, x_{i2}, \ldots, x_{iM})^{\top}$ denote the response vector. For question $j$, let $\boldsymbol{x}_j^{(e)} = (x_{1j}, x_{2j}, \ldots, x_{Nj})^{\top}$ denote its response vector. Here, $x_{ij}, i = 1, \ldots, N, j = 1 \ldots, M$ is obtained as follows:

$$x_{ij} = \begin{cases} 1, & \text{if } r_{ij} = 1, \\ 0, & \text{if } (s_i, e_j, 0) \notin \mathcal{D} \text{ and } (s_i, e_j, 1) \notin \mathcal{D}, \\ -1, & \text{otherwise.} \end{cases} \quad (4)$$

Next, ID-CDF utilizes a learner diagnostic function $\mathcal{F}(\cdot)$ and a question diagnostic function $\mathcal{G}(\cdot)$ to diagnose learner traits and

question parameters respectively, as shown in the following:

$$\boldsymbol{\theta}_i = \mathcal{F}\left(\boldsymbol{x}_i^{(s)}; \omega^{(s)}\right), \quad i = 1, 2, \ldots, N \quad (5)$$

$$\boldsymbol{\psi}_j = \mathcal{G}\left(\boldsymbol{x}_j^{(e)}; \omega^{(e)}\right), \quad j = 1, 2, \ldots, M, \quad (6)$$

where $\boldsymbol{\theta}_i$ denotes learner traits, and $\boldsymbol{\psi}_j$ denotes question parameters. All $\omega^{(\cdot)}$ denote latent parameters of diagnostic functions that reflect the diagnostic process and can be learned from data.

Next, we give the formal definition of the indispensable identifiability condition and monotonicity condition of the diagnostic functions.

*Definition 3.5.* **Identifiability Condition.** A diagnostic function satisfies the identifiability condition if and only if diagnostic results are completely mined from observable response data, and there does not exist any exterior unobservable factor that affects the calculation of diagnostic results.

The identifiability condition of diagnostic functions satisfies the identifiability of diagnostic results through its **contrapositive**. Specifically, ID-CDF regularizes the one-to-one map relationship between diagnostic results and observable response data. Let $r$ be the actual distribution of a learner's response data which can be vectorized as $x^{(s)}$. Let $\tilde{r}$ be another learner's response data distribution such that $\tilde{r} = r$ and can be vectorized as $\tilde{x}^{(s)}$. Because the diagnostic function $\mathcal{F}\left(\cdot; \omega^{(s)}\right)$ satisfies the identifiability condition, we can get $\mathcal{F}\left(\tilde{x}^{(s)}; \omega^{(s)}\right) = \mathcal{F}\left(x^{(s)}; \omega^{(s)}\right)$, i.e., $\tilde{\boldsymbol{\theta}} = \boldsymbol{\theta}$. As a result, given the identifiability condition, we can get:

$$\tilde{r} = r \rightarrow \tilde{\boldsymbol{\theta}} = \boldsymbol{\theta}, \quad (7)$$

which is indeed the **contrapositive** of the learner identifiability defined in Eq.(1), and logically equivalent to it. As a result, the identifiability of learner traits is equivalently satisfied. The identifiability of question parameters is satisfied in the same way.

*Definition 3.6.* **Monotonicity Condition.** For any learner diagnostic function $\mathcal{F} : R \rightarrow \Theta$, the function satisfies the monotonicity condition if and only if it is monotonically increasing at any dimension of response vectors, i.e., $\frac{\partial \mathcal{F}}{\partial x_{ij}^{(s)}} \geq 0, \forall j = 1, 2, \ldots, M$.

The monotonicity condition addresses the explainability overfitting problem from the perspective of inductive learning. Specifically,

ID-CDF applies the condition to the diagnostic module, which is shared by *all learners*. With its help, the diagnostic module can *directly* generate monotonic learner traits from response score distribution, rather than learn them from parameter optimization given single response scores. Considering the distribution consistency of observed and unobserved response data, the diagnostic module can also achieve similar explainability on the latter as on the former, which addresses the explainability overfitting problem.

**Predictive Module**. The predictive module aims to reconstruct response scores from learner traits and question parameters to ensure the preciseness of diagnostic results. In ID-CDF, the predictive module consists of pre-given knowledge factors (Q-matrix in Figure 4 that specifies the mapping between questions and knowledge concepts), and a flexible interaction function that models the complex interaction between learners and questions. The reconstruction process is defined in the following:

$$y_{ij} = \mathcal{H}\left(\boldsymbol{\theta}_i \odot \boldsymbol{q}_j, \boldsymbol{\psi}_j \odot \boldsymbol{q}_j; \omega^{(p)}\right), \quad (8)$$

where $\mathcal{H}$ denotes the interaction function. The $q_j$ denotes the binary vector of question $j$ in the Q-matrix which indicates the required knowledge concepts of the question. The $\odot$ denotes element-wise product, which is used to mask irrelevant knowledge concepts in the training of ID-CDF, inspired by [7, 31]. The $\omega^{(p)}$ denotes learnable latent parameters of $\mathcal{H}(\cdot)$ that models the complex interaction between learners and questions. In ID-CDF, the interaction function is flexible and can be integrated into various forms, which depends on the actual demand for web learner modeling.

**Loss Function**. In CD-based learner modeling, response scores are usually binary. As a result, we utilize the cross entropy between the output $y$ and true score $r$ as the loss function of ID-CDF, as shown in Eq (9):

$$\mathcal{L}(\Omega) = -\sum_{(s_i, e_j, r_{ij}) \in \mathcal{D}} \left(r_{ij} \log y_{ij} + (1 - r_{ij}) \log(1 - y_{ij})\right), \quad (9)$$

where $\Omega = \left(\omega^{(s)}, \omega^{(e)}, \omega^{(p)}\right)$ denotes the parameters of ID-CDF.

## 3.3 ID-CDM: An Implementation of ID-CDF

We present an Identifiable Cognitive Diagnosis Model (ID-CDM) as an implementation of ID-CDF. It consists of a neural network-based diagnostic module and a predictive module, which utilizes the powerful representation learning capability of neural networks to capture cognitive states and question parameters from data.

**Diagnostic Module**. The principle of the design of the diagnostic module is to satisfy the identifiability condition and the monotonicity condition while ensuring diagnosis preciseness. To this end, in ID-CDM, we adopt multi-layer perceptrons (MLPs) with parameter constraints to learn the diagnosis process from data, while keeping the two pivotal conditions. Specifically, the learner diagnostic module is defined as follows:

$$f_1 = \sigma(W_1^{(s)} \times \boldsymbol{x}_i^{(s)} + b_1^{(s)}), \quad (10)$$

$$\boldsymbol{\theta}_i = \sigma(W_2^{(s)} \times f_1 + b_2^{(s)}), \quad (11)$$

where $\sigma(\cdot)$ denotes the sigmoid activation function. In this learner diagnostic module, the learnable parameters can be represented as $\omega^{(s)} = \left(W_1^{(s)}, W_2^{(s)}, b_1^{(s)}, b_2^{(s)}\right)$. The $W_1^{(s)}$ and $W_2^{(s)}$ are constrained to be positive to satisfy the monotonicity condition. Since there does

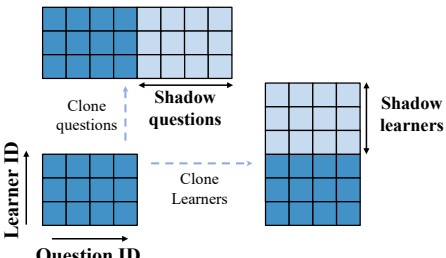

**Figure 5: An illustration of data augmentation in RQ1.**

not exist any exterior unobservable factors, this module satisfies the identifiability condition.

Meanwhile, the question diagnostic module is defined as:

$$g_1 = \sigma(W_1^{(e)} \times \boldsymbol{x}_j^{(e)} + b_1^{(e)}), \quad (12)$$

$$g_2 = \sigma(W_2^{(e)} \times g_1 + b_2^{(e)}), \quad (13)$$

$$\boldsymbol{\psi}_j = \sigma(W_3^{(e)} \times g_2 + b_3^{(e)}), \quad (14)$$

where the learnable parameters can be represented as $\omega^{(e)} = \left(W_1^{(e)}, W_2^{(e)}, W_3^{(e)}, b_1^{(e)}, b_2^{(e)}, b_3^{(e)}\right)$. Because there do not exist any exterior unobservable factors, this module also satisfies the identifiability condition.

**Predictive Module**. In the predictive module, we also adopt neural networks to learn the complex interaction between learners and questions. Specifically, we first utilize single-layer perceptrons to aggregate knowledge concept-wise diagnostic results to low-dimensional features to gain more effective representations of learners and questions. Next, we utilize an MLP to reconstruct response scores from aggregated representations.

To begin with, the aggregation layer of diagnostic output is defined as follows:

$$\boldsymbol{\alpha}_i = \sigma(W^{(u)} \times (\boldsymbol{\theta}_i \odot \boldsymbol{q}_j) + b^{(u)}), \quad (15)$$

$$\boldsymbol{\phi}_j = \sigma(W^{(v)} \times (\boldsymbol{\psi}_j \odot \boldsymbol{q}_j) + b^{(v)}). \quad (16)$$

Next, aggregated representations of learner $s_i$ and question $e_j$ are input to a three-layer MLP to reconstruct response scores:

$$z_1 = \sigma(W_1^{(c)} \times (\boldsymbol{\alpha}_i - \boldsymbol{\phi}_j) + b_1^{(c)}), \quad (17)$$

$$z_2 = \sigma(W_2^{(c)} \times z_1 + b_2^{(c)}), \quad (18)$$

$$y_{ij} = \sigma(W_3^{(c)} \times z_2 + b_3^{(c)}). \quad (19)$$

In the predictive module, parameters can be represented as $\omega^{(p)} = \left(W^{(u)}, W^{(v)}, W_1^{(c)}, W_2^{(c)}, W_3^{(c)}, b^{(u)}, b^{(v)}, b_1^{(c)}, b_2^{(c)}, b_3^{(c)}\right)$. These parameters can be learned together with $\omega^{(s)}$ and $\omega^{(e)}$ in the training of ID-CDM.

## 4 EXPERIMENT

## 4.1 Experiment Overview

In this section, we conduct experiments on four real-world datasets to demonstrate the identifiability, explainability and preciseness of ID-CDF[3]. The experiments aim to answer four research questions in the following:

---

[3]Our code is available at https://anonymous.4open.science/r/ID-CDF-AB86/.

**Table 1: Identifiability Score (IDS ↑) of diagnostic results of CDMs (RQ1). $\mathcal{I}(X)$ indicates whether $X$ is identifiable.**

| CDM | IDS ↑ of Learner Diagnostic Result $\Theta$ | | | | | IDS ↑ of Question Diagnostic Result $\Psi$ | | | | |
|---|---|---|---|---|---|---|---|---|---|---|
| | ASSIST | Algebra | Math1 | Math2 | $\mathcal{I}(\Theta)$ | ASSIST | Algebra | Math1 | Math2 | $\mathcal{I}(\Psi)$ |
| DINA | 0.550±0.003 | 0.092±0.004 | 0.451±0.006 | 0.368±0.009 | ✗ | 0.208±0.001 | 0.160±0.000 | 0.193±0.019 | 0.214±0.038 | ✗ |
| IRT | 0.691±0.004 | 0.698±0.005 | 0.690±0.004 | 0.688±0.003 | ✗ | 0.376±0.001 | 0.371±0.000 | 0.543±0.035 | 0.540±0.036 | ✗ |
| MIRT | 0.047±0.001 | 0.045±0.000 | 0.046±0.000 | 0.047±0.001 | ✗ | 0.041±0.000 | 0.042±0.000 | 0.085±0.005 | 0.076±0.006 | ✗ |
| NCDM | 0.857±0.001 | 0.409±0.001 | 0.662±0.005 | 0.597±0.006 | ✗ | 0.616±0.000 | 0.480±0.000 | 0.420±0.012 | 0.307±0.009 | ✗ |
| NCDM-Const | 0.897±0.001 | 0.701±0.005 | 0.688±0.003 | 0.635±0.006 | ✗ | 0.968±0.000 | 0.989±0.000 | 0.915±0.010 | 0.916±0.009 | ✗ |
| CDMFKC | 0.621±0.001 | 0.390±0.001 | 0.613±0.015 | 0.553±0.011 | ✗ | 0.618±0.000 | 0.481±0.000 | 0.408±0.012 | 0.297±0.011 | ✗ |
| ID-CDM-nEnc | 0.613±0.001 | 0.375±0.002 | 0.595±0.008 | 0.524±0.026 | ✗ | 0.601±0.000 | 0.495±0.000 | 0.401±0.007 | 0.304±0.008 | ✗ |
| ID-CDM | **1.000**±0.000 | **1.000**±0.000 | **1.000**±0.000 | **1.000**±0.000 | ✔ | **1.000**±0.000 | **1.000**±0.000 | **1.000**±0.000 | **1.000**±0.000 | ✔ |

- **RQ1**: How is the identifiability of diagnostic results of ID-CDM?
- **RQ2**: How is the explainability of diagnostic results of ID-CDM?
- **RQ3**: Can diagnostic results of ID-CDM accurately reflect learners' response performances?
- **RQ4**: How is the statistical relationship between the diagnosed learner traits and actual learner performance?

### 4.2 Experimental Setup

**Dataset description**. In experiments, we utilize four public real-world datasets with different characteristics, including two online K-12 mathematical test datasets collected from online learning platforms, i.e., ASSIST (ASSISTments 2009-2010 "skill builder") [8] and Algebra (Algebra | 2006-2007) [16], and two offline high school mathematical exam datasets, i.e., Math1 and Math2 [22]. A summary of the datasets is available in Appendix A.2. In the preprocessing of datasets, for online test datasets, we reserve only the first attempt of learners answering a question. To ensure that each learner has enough response data for personalized modeling, we remove learners with less than 15 response logs. For the Algebra dataset, we randomly selected 100,000 questions for our experiment. Next, 80% of each learner's response log is randomly split as a train set, while the rest 20% serves as the test set. In the train set, 90% of each learner's response log is used for model training, while the rest 10% is used for model validation.

**Baselines**. We compare the performance of ID-CDM with five typical score prediction-based CDMs and two encoder-decoder models in our experiment. These baselines are described as follows.

- **DINA** [7] is a score prediction-based CDM that models learner abilities as binary knowledge proficiencies, and models question parameters by 'guess' and 'slip' probabilities.
- **IRT** [2] is a score prediction-based CDM that models scalar learner abilities, question difficulties and question discrimination through a logistic-like interaction function.
- **MIRT** [27] is a score prediction-based CDM that extends scalar learner abilities and question difficulties in IRT to multidimensional situations.
- **NCDM** [31] is a score prediction-based CDM. NCDM utilizes a monotonic neural network to learn the complex interaction between learners and questions and can diagnose knowledge concept-wise learner abilities and question difficulties.
- **CDMFKC** [20] is a score prediction-based CDM that utilizes an elaborately designed neural network to model the influence of knowledge concepts on learners' learning performance.

- **U-AutoRec** [28] is an encoder-decoder model which utilizes an autoencoder to learn user traits from historical response logs.
- **CDAE** [35] is an encoder-decoder model that utilizes a denoising autoencoder to facilitate robustness in learning user traits from historical response logs.

**Training setting**. In the training setting, we implement all models with PyTorch using Python. The dimension of diagnostic results of MIRT is set to 16. The dimension of transformed diagnostic results of ID-CDM is set to 64. The dimension of U-AutoRec is set to the number of knowledge concepts so that we can explore in RQ2 the explainability of traditional encoder-decoder models to validate their usability in CD-based learner modeling. All model parameters are initialized with Xavier normal method [14], and optimized with the Adam algorithm [18]. All experiments are run on a Linux server with two 2.10GHz Intel Xeon E5-2620 v4 CPUs and one NVIDIA Tesla-A100 GPU.

### 4.3 Identifiability Evaluation (RQ1)

In this part, we design a novel experiment to quantitatively evaluate the identifiability of various CDMs. Our motivation is that the identifiability of CDMs can be evaluated by measuring the distance between traits of learners/questions with the same response distribution. If diagnostic results are identifiable, the distance between them should be close to zero.

There are two challenges in achieving this goal. The first challenge is the lack of learners/questions with the same response distribution in real data because of data sparsity. The second challenge is the lack of quantitative evaluation metrics of identifiability. For the first one, inspired by data augmentation in computer vision and natural language processing [1, 13], we propose a data augmentation operation that copies the response score matrix to obtain "shadow" learners/questions that have the same response score distribution with the original, as shown in Figure 5. For the second one, to quantitatively validate the identifiability of CDMs on the augmented data, we propose the *Identifiability Score* (IDS) as an indicator of the identifiability. The more similar diagnostic results of original and shadow learners/questions, the larger the value of IDS. In addition, the full score of IDS denotes rigorous identifiability. To achieve this goal, we define IDS of learner traits $\Theta$ as follows:

$$IDS(\Theta) = \frac{1}{Z} \sum_{i \in S} \sum_{j \in S} \frac{I(\boldsymbol{r}_i = \boldsymbol{r}_j) \wedge I(i \neq j)}{\left[1 + dist(\boldsymbol{\theta}_i, \boldsymbol{\theta}_j)\right]^2}, \qquad (20)$$

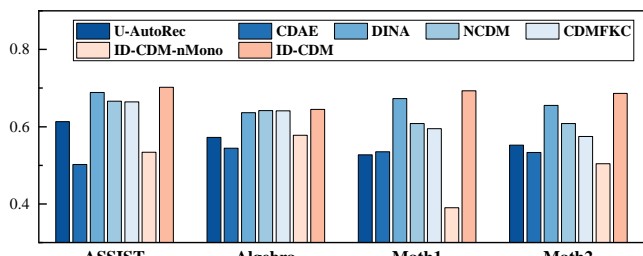

(a) Degree of Consistency ($\overline{DOC}$) ↑ of CDMs and encoder-decoder models.

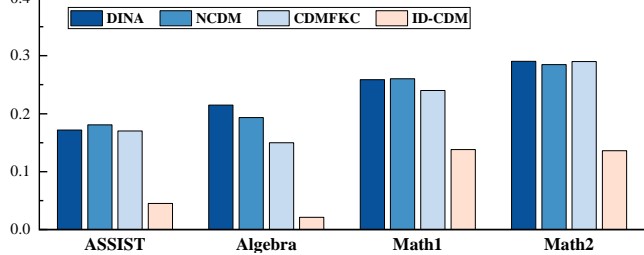

(b) Rate of Explainability Overfitting ($REO$) ↓ of CDMs.

**Figure 6: Results of the explainability of diagnosed learner traits (RQ2).**

where $Z = \sum_{i \in S} \sum_{j \in S} I(\mathbf{r}_i = \mathbf{r}_j) \wedge I(i \neq j)$. The $dist(\boldsymbol{\theta}_i, \boldsymbol{\theta}_j)$ is the Manhattan distance [6] between learner $i$'s traits and learner $j$'s traits. As mentioned above, $IDS(\Theta)$ is monotonically decreasing at $dist(\boldsymbol{\theta}_i, \boldsymbol{\theta}_j)$. **Learner traits are rigorously identifiable if and only if** $IDS(\Theta) = 1$. Similarly, we can also evaluate the identifiability of question parameters $\Psi$ by calculating $IDS(\Psi)$.

We evaluate the identifiability of score prediction-based CDMs and ID-CDM. Furthermore, we also explore the impact of random initialization in existing CDMs and diagnostic modules of ID-CDF on the identifiability by an **ablation study**. For the impact of random initialization in existing CDMs, we initialize diagnostic results of NCDM by constant values (namely NCDM-Const) and compare its IDS with that of the original NCDM. For the impact of the diagnostic module of ID-CDF, we remove diagnostic modules of ID-CDM (namely ID-CDM-nEnc) and compare its IDS with that of the original ID-CDM. The experimental results are presented in Table 1. Within score prediction-based CDMs, none of them can generate identifiable diagnostic results due to the randomness existing in the optimization. Indeed, the improvement of the IDS of NCDM-Const relative to NCDM demonstrates the impact of random initialization on the identifiability. However, such an improvement is limited, and diagnostic results of NCDM-Const are still unidentifiable. On the other hand, the IDS of ID-CDM always reaches the maximum value (i.e., IDS = 1), which means that learner traits diagnosed by ID-CDM are rigorously identifiable. This result is in alignment with our analysis in Section 3.2. Moreover, comparing the IDS of ID-CDM and that of ID-CDM-nEnc, we can deduce that the diagnostic module plays an indispensable role in guaranteeing identifiability. In conclusion, our proposal can effectively address the non-identifiability problem in CD-based learner modeling.

### 4.4 Explainability Evaluation (RQ2)

In this part, we evaluate the explainability of CDMs from the aspect of monotonicity assumption. Furthermore, we also quantitatively evaluate the explainability overfitting problem of score prediction-based CDMs mentioned above. Our motivation is that the order of explainable learners' knowledge proficiencies should be consistent with the order of response scores on relevant questions. To this end, inspired by previous works [10], we propose the *Degree of Consistency* (DOC) as the evaluation metric. Given question $e_l, l = 1, 2, \ldots, M$, $DOC$ is defined as follows:

$$DOC(e_l) = \frac{\sum_{i,j} \delta(r_{il}, r_{jl}) \sum_{k=1}^{K} q_{lk} \wedge J(l, i, j) \wedge \delta(\theta_{ik}, \theta_{jk})}{\sum_{i,j} \delta(r_{il}, r_{jl}) \sum_{k=1}^{K} q_{lk} \wedge J(l, i, j) \wedge I(\theta_{ik} \neq \theta_{jk})}, \quad (21)$$

where $\delta(x, y) = 1$ if $x > y$ and $\delta(x, y) = 0$ otherwise. $J(l, i, j) = 1$ if both $s_i$ and $s_j$ has answered question $e_l$ and $J(l, i, j) = 0$ otherwise. $I(\cdot)$ denotes the indicator function. The $DOC$ is in $[0, 1]$. **The higher the DOC, the better the explainability of diagnosed learner traits.** Next, we calculate the average DOC as the measurement of the explainability of learner traits, i.e., $\overline{DOC} = \frac{1}{M} \sum_{l=1}^{M} DOC(e_l)$. Next, to explore the explainability overfitting problem of CDMs, we aim to compare $\overline{DOC}$ on test data with that on training data. To this end, we propose the Rate of Explainability Overfitting ($REO$) to measure the discrepancy between them. The $REO$ is defined as follows:

$$REO(\mathcal{D}_{train}, \mathcal{D}_{test}) = 1 - \frac{\overline{DOC}(\mathcal{D}_{test})}{\overline{DOC}(\mathcal{D}_{train})}, \quad (22)$$

where $\mathcal{D}_{train}, \mathcal{D}_{test}$ denotes training data and test data respectively. The $REO$ indeed evaluates the rate of discrepancy between $\overline{DOC}(\mathcal{D}_{test})$ and $\overline{DOC}(\mathcal{D}_{train})$ to $\overline{DOC}(\mathcal{D}_{train})$. The $REO$ is generally in $[0, 1]$. **The smaller the $REO$, the more smaller the explainability metric $\overline{DOC}$ in the training and test dataset.**

Experimental results are shown in Figure 6. We evaluate the explainability of score prediction-based CDMs (DINA, NCDM, CDM-FKC), encoder-decoder models (U-AutoRec, CDAE), and ID-CDM. We also conduct an **ablation study** where we remove the monotonicity condition of ID-CDM to get ID-CDM-nMono to explore the impact of the monotonicity condition on the explainability of ID-CDM. IRT and MIRT are excluded from this experiment because they cannot generate knowledge concept-wise learner traits. We further evaluate the explainability overfitting of CDMs (DINA, NCDM, CDMFKC, ID-CDM). Encoder-decoder models are excluded from this experiment because they have been experimentally validated to be less explainable in the left part of the figure. From Figure 6, we first observe that the $\overline{DOC}$ of encoder-decoder models is always lower than that of CDMs, which means that traditional encoder-decoder models are incapable of diagnosing explainable learner traits. On the other hand, the $\overline{DOC}$ of ID-CDM is always higher than that of baselines, which illustrates that ID-CDM has the state-of-the-art explainability of learner traits. In addition, the $\overline{DOC}$ of ID-CDM-nMono is much lower than that of ID-CDM in all cases. This observation demonstrates the decisive impact of the monotonicity condition of ID-CDM on the explainability of learner traits. Next, we can observe that the $REO$ of ID-CDM is significantly lower than other baseline CDMs, which means that the discrepancy between the explainability of ID-CDM on test data and training

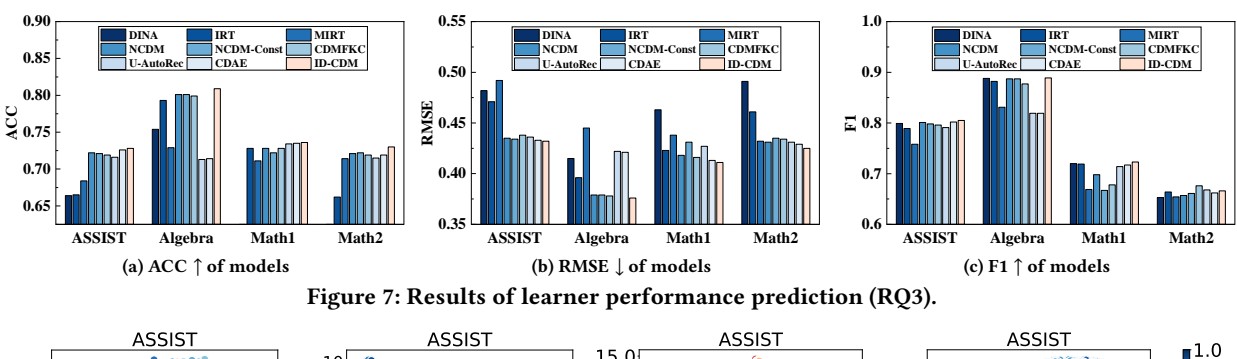

Figure 7: Results of learner performance prediction (RQ3).

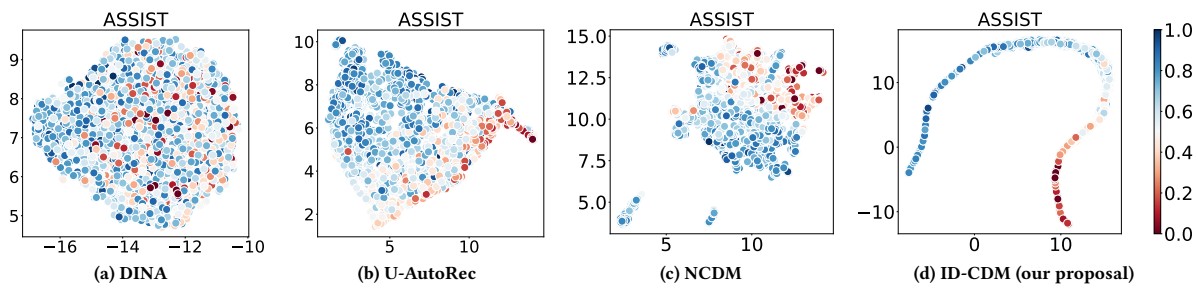

Figure 8: Clustering of learner traits diagnosed by different models (RQ4). Points are colored with correct rates.

data is much smaller than that of baselines. In conclusion, ID-CDM can effectively alleviate the explainability overfitting of CDMs.

## 4.5 Learner Score Prediction (RQ3)

In CD-based learner modeling, it is hard to explicitly evaluate diagnosis preciseness because learners' real cognitive states are unobservable. A common solution is to evaluate the response score prediction performance of CDMs to assess diagnosis preciseness implicitly. To this end, we evaluate the score prediction performance of different models from the aspect of both classification and regression. We utilize Accuracy (ACC), F1-score (F1), and Rooted Mean Square Error (RMSE) as the evaluation metrics. The classification threshold is 0.5. To guarantee fairness, we utilize the diagnostic output of ID-CDM from the training data rather than the test data to predict learners' performance in the test data.

The experimental results are shown in Figure 7. We can observe that the performance of ID-CDM on learner score prediction exceeds the performance of baselines in most cases. Actually, ID-CDM utilizes neural networks to learn the complicated diagnostic process, it can also capture the complex interaction between learners and questions similar to NCDM and CDMFKC. In conclusion, ID-CDM has a competitive diagnosis preciseness while ensuring the identifiability and explainability of diagnostic results.

## 4.6 Learner traits clustering (RQ4)

To explore the statistical relationship between diagnosed learner traits and learners' actual performance, we visualize learner diagnostic results of CDMs by UMAP [23] and color points of learners by their correct rates. Then we explore whether diagnosed learner traits can distinguish between learners with different correct rates. The experimental result[4] in ASSIST is shown in Figure 8. We can observe from the figure that using diagnostic results of baseline models, it is hard to distinguish between learners with high correct rates and those with low correct rates. On the other hand, the shape

---

[4]Complete results are available in Appendix A.4.

of learner traits diagnosed by ID-CDM is ribbon-like and is consistent with the direction of the changing of learners' correct rates. So we can easily distinguish learners with different correct rates from diagnostic results of ID-CDM. As a result, there is a strong correlation between the learner traits and their actual performance. This result further demonstrates the explainability of ID-CDM on response data from the perspective of statistics and visualization.

## 5 CONCLUSION

In this paper, we studied the non-identifiability and explainability overfitting problems that widely exist in the CD-based learner modeling task and proposed an identifiable cognitive diagnosis framework (ID-CDF) to address the two issues. Specifically, we first proposed a novel response-proficiency-response (R-P-R) paradigm to address the two problems from their roots. Based on this paradigm, we proposed ID-CDF, which utilizes inductive diagnostic modules to obtain identifiable and explainable diagnostic results from observed response data. It then uses a predictive module that models the complex interaction between learners and questions to guarantee the preciseness of diagnostic results. We then proposed ID-CDM as an implementation of ID-CDF to show its usability. Finally, we demonstrated the effectiveness of ID-CDF through extensive experiments on four real-world datasets.

This work also opens some future research directions. First, people can further mine the potential of the diagnostic module and introduce various types of behavioral data to CD-based learner modeling. Second, in web learning scenarios, learners' cognitive states might change over time [41, 43], which is different from the stationary cognitive states in traditional cognitive diagnosis. Along this line, people can improve the design of CDMs based on our proposed paradigm to capture the dynamic change of cognitive states and improve the usability of CD-based personalized modeling techniques. In a word, we hope this work can inspire more exploration in personalized learner modeling research in the future.

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

**Table 2: Dataset summary.**

| Statistics | ASSIST | Algebra | Math1 | Math2 |
|---|---|---|---|---|
| # Learners | 4,163 | 1,336 | 4,209 | 3,911 |
| # Questions | 17,746 | 100,000 | 20 | 20 |
| # Knowledge concepts | 123 | 491 | 11 | 16 |
| # Response logs | 324,572 | 322,808 | 84,180 | 78,220 |
| # KC[1] per question | 1.19 | 1.12 | 3.35 | 3.20 |
| # Answers per learner | 107.26 | 259.49 | 20.0 | 20.0 |
| Correct rate | 0.654 | 0.795 | 0.424 | 0.415 |

[1]"KC" denotes knowledge concepts.

# A APPENDIX

## A.1 The Computational Complexity of ID-CDM

The computational complexity of ID-CDM consists of two parts, i.e., diagnosis complexity $T_{diag}$ and prediction complexity $T_{pred}$.

**Diagnosis Complexity**. Given $N$ learners, $M$ questions, $K$ knowledge concepts, $H$ hidden layers, and the dimension of a hidden layer as $D$. Then there are $O(N \cdot D + M \cdot D)$ calculations in input layers, $O(H \cdot D^2)$ in hidden layers, and $O(D \cdot K)$ calculations in output layers. Given a pair of learner response logs and question response logs, the diagnosis computational complexity is $T_{diag} = O((N + M) \cdot D + H \cdot D^2 + 2 \cdot D \cdot K)$.

**Prediction Complexity**. Given conditions above, the predictive module of ID-CDM first aggregetes diagnostic results to low dimensional representations by single-layer perceptrons, where the computational complexity is $O(K \cdot D)$. Then the aggregated representations are input to a MLP to reconstruct the response score, where the computational complexity is $O(H \cdot D^2)$. As a result, the prediction computational complexity is $T_{pred} = O(K \cdot D + H \cdot D^2)$.

## A.2 A Summary of Datasets

A summary of the four real-world datasets is shown in Table 2.

## A.3 Case Study

A case study in Math1 is shown in Figure 9. In the right part of the figure, NCDM (Shadow) denotes diagnostic results of NCDM on the shadow learner with the same response score distribution. We have several analysis of the diagnostic results. First, both the diagnostic results of ID-CDM and NCDM are explainable. It should be noticed that the order of correct rates is not rigorously consistent with the order of knowledge proficiencies, because the difficulty of questions on different knowledge concepts are not exactly equal. Second, the diagnostic result of ID-CDM is more discriminated than that of NCDM. The range of diagnostic results of ID-CDM is approximately from 0.5 to 1.0, while that of NCDM is approximately from the 0.4 to 0.6. The low variance of NCDM is consistent with the observation in [31]. Third, the non-identifiability of NCDM. Comparing NCDM and NCDM (Shadow), their diagnostic results are not equal.

## A.4 Learner Traits Clustering

Detailed learner traits clustering results are shown in Figure 10, Figure 11, Figure 12 and Figure 13. A summarization of the statistical features of these learner traits is listed as follows.

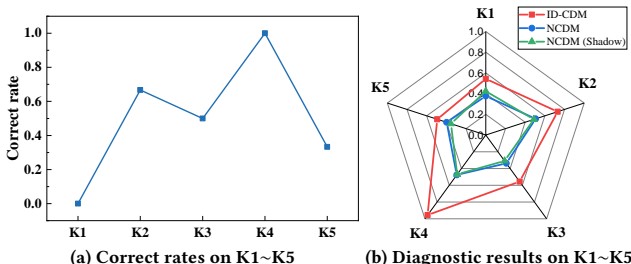

(a) Correct rates on K1~K5      (b) Diagnostic results on K1~K5

**Figure 9: Learner diagnostic results in Math1.**

**ID-CDM and ID-CDM-nMono (Figure 10)**. In ID-CDM and ID-CDM-nMono, the distribution of learner traits is highly correlated with the distribution of correct rates. If we view correct rates as labels (0.5 as the threshold), then learners are linearly separable. Furthermore, comparing the result of ID-CDM and ID-CDM-nMono, the monotonicity condition actually tightens the distribution of learner traits in the orthogonal direction of the changing of correct rates, which enhances the correlation between the distribution of learner traits and correct rates.

**NCDM and CDMFKC (Figure 11)**. In NCDM and CDMFKC, the distribution of learner traits is partially correlated with the distribution of correct rates. In Math1 and Math2 dataset, learners are also linearly separatable if we view correct rates as labels. However, the shape of the distribution of learner traits in the four datasets is irrelevant to the learner traits. Moreover, in ASSIST and Algebra, learner traits are not linearly separable, and some points with negative labels (i.e., correct rate < 0.5) have been mixed with others with positive labels. Comparing Figure 11 and Figure 10, we conclude that the diagnostic module of ID-CDF enables CDMs to learn the correlation between the shape of the distribution of learner traits and the distribution of correct rates, which enhances the discrimination ability of CDMs.

**DINA and MIRT (Figure 12)**. In DINA and MIRT, the distribution of learner traits is almost uncorrelated with the distribution of correct rates except for results of DINA in Math1 and Math2. Although DINA can diagnose knowledge concept-wise learner traits, and its logistic-like interaction function is intrinsically explainable, the dichotomy of learner traits limits the ability of DINA to capture the correlation between learner traits and response patterns in large-scale data that consists of hundreds of knowledge concepts and hundreds of thousands of response logs, such as ASSIST and Algebra. As for MIRT, the distribution of learner traits is irrelevant to correct rates because MIRT models learners by low-dimensional latent traits whose components are not knowledge concept-wise thus lack explainability.

**U-AutoRec and CDAE (Figure 13)**. In U-AutoRec and CDAE, the distribution of learner traits is partially correlated with the distribution of correct rates. However, compared to results of ID-CDM, learner traits of U-AutoRec and CDAE are not linearly separatable, and the shape of these distributions are not well correlated with correct rates. These results indicate that the seperately designed learner and question diagnostic modules and the monotonicity condition of ID-CDM effectively facilitates its ability to capture the correlation between learner traits and correct rates, which actually makes diagnostic results of ID-CDM more feasible.

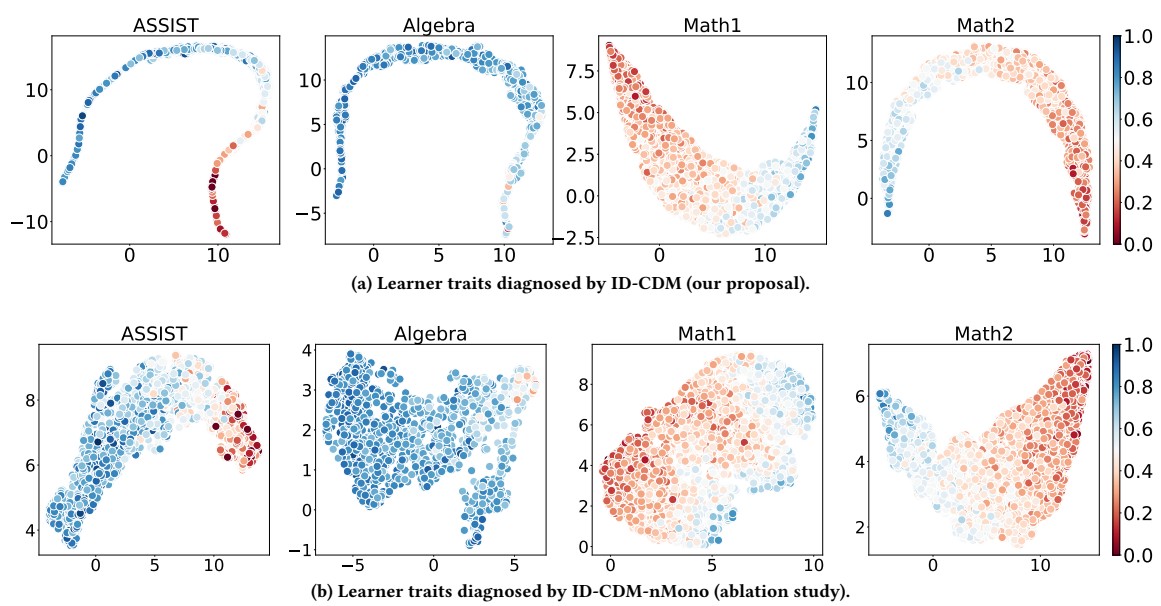

Figure 10: Learner traits clustering (Part 1). Each point denotes a learner's traits, colored by his/her correct rate.

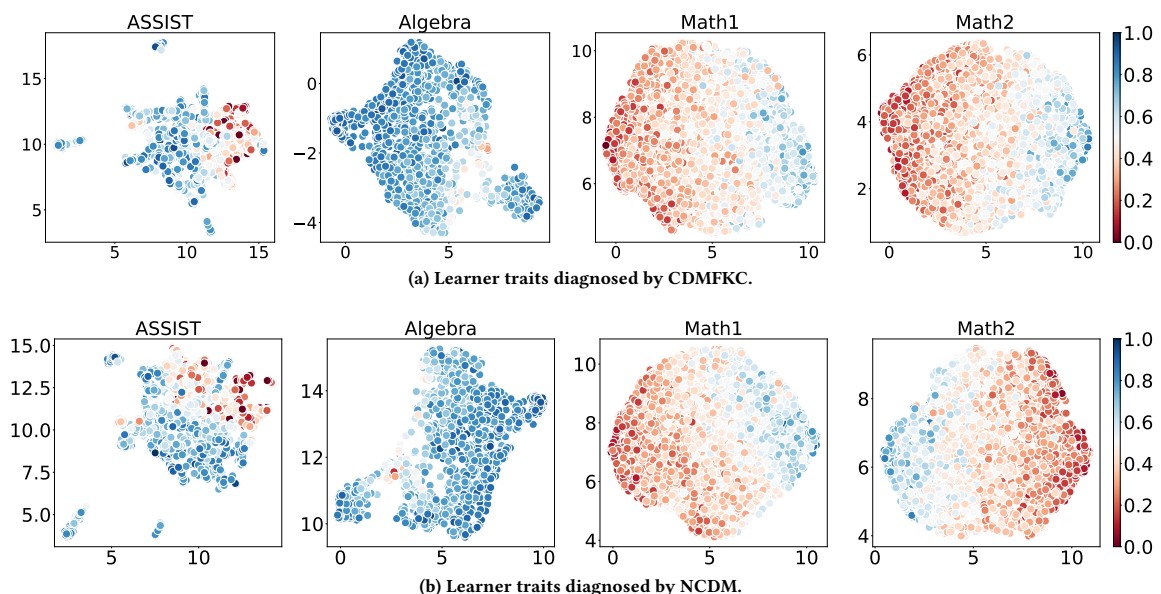

Figure 11: Learner traits clustering (Part 2). Each point denotes a learner's traits, colored by his/her correct rate.

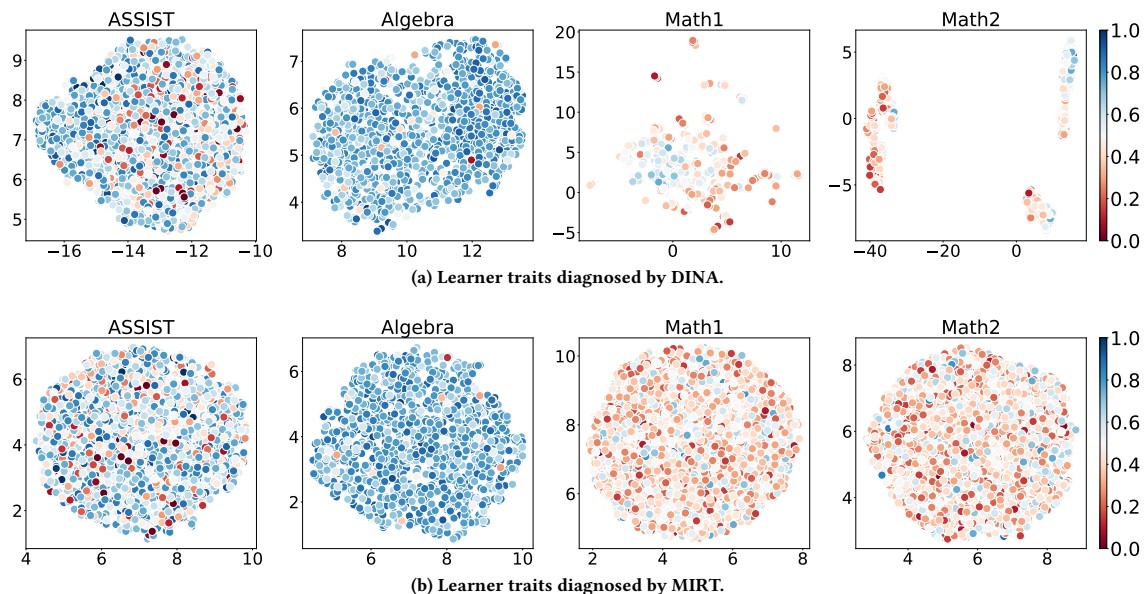

Figure 12: Learner traits clustering (Part 3). Each point denotes a learner's traits, colored by his/her correct rate.

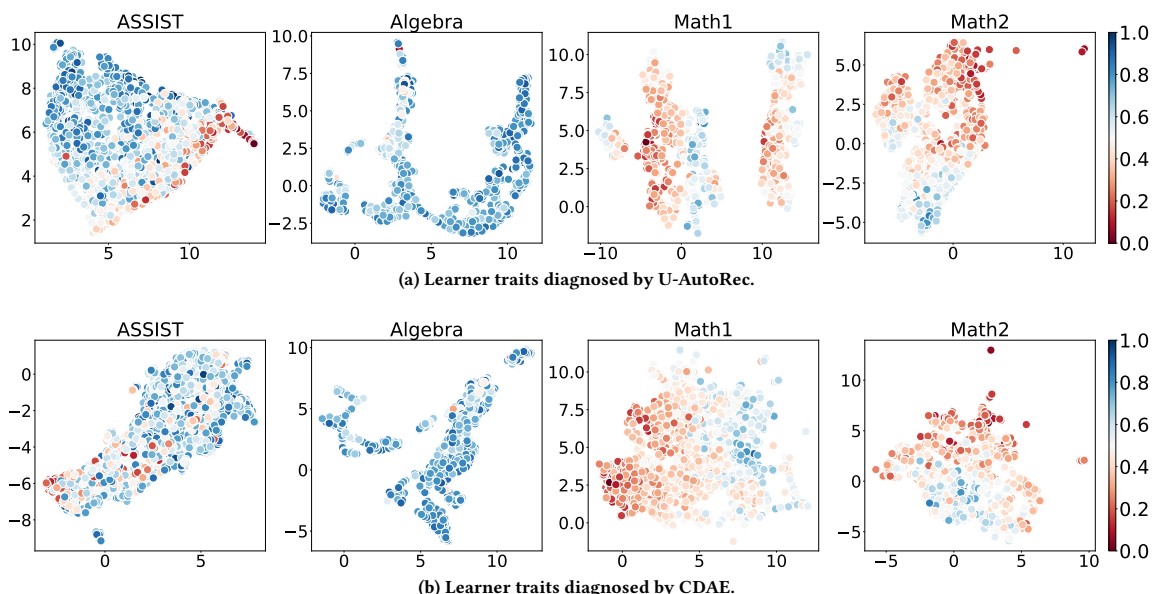

Figure 13: Learner traits clustering (Part 4). Each point denotes a learner's traits, colored by his/her correct rate.

