# OpenReview forum: "Towards the Identifiability and Explainability for Personalized Learner Modeling: An Inductive Paradigm"
_ACM.org/TheWebConf/2024/Conference — TheWebConf24_

### Official Review · Reviewer_TCfD · 2023-11-19

**Novelty:** 3
**Technical Quality:** 3

**Review:**

This paper proposes improved approach to identify latent user knowledge states and predict user response to knowledge testing questions. The approach provides better identifiability and explainability compared to existing benchmarks

Pros
1. Novel architecture that provides required properties of estimated parameters and in the same time gives same of better prediction accuracy

Cons
1. Application area is very specific to user knowledge states modeling in education

**Questions:**

1. Could you share accuracy comparison results as a table rather than as a chart
2. What is impact of additional components that improves identifiability and interpretability on prediction accuracy
3. How does your approach account for the dynamic student knowledge states trajectories over time?
4. How did control for potential risk of overfitting in your inductive models?

**Ethics Review Description:**

no concerbs

**Reviewer Confidence:**

2: The reviewer is willing to defend the evaluation, but it is likely that the reviewer did not understand parts of the paper

**Scope:**

3: The work is somewhat relevant to the Web and to the track, and is of narrow interest to a sub-community

---

### Official Review · Reviewer_HB8w · 2023-11-20

**Novelty:** 4
**Technical Quality:** 4

**Review:**

Summary:

This paper aims to improve the identifiability and explainability of cognitive diagnosis based personalized learner modeling methods. Specifically, this paper propose a response-proficiency-response paradigm to address the two problems mentioned above. Furthermore, based on the proposed paradigm, this paper introduces a new framework, ID-CDF, which utilizes inductive diagnostic modules to obtain identifiable and explainable diagnostic results. With the improved experimental results on four publicly available datasets, the authors concluded that the proposed method can achieve the state-of-the-art performance.


Strong points:

1)	The studied problem (i.e., personalized learner modeling) is interesting and promising, with many applications in online learning platforms.
2)	The authors conducted extensive experiments and the experimental results is promising.
3)	The figures in this paper are drawn very beautifully and meticulously.


Weak points:

1)	In line 90 of the first page, the authors mention 'through our investigation,' but this paper does not provide specific details of this investigation, leading to a lack of foundation for that part of the discussion. Please could the authors complete the specific content of this investigation.
2)	I don't quite agree with the summary of the first contribution in this paper. I don't agree that the authors 'discovered the non-identifiability and explainability overfitting problem', but rather just elaborated on these two issues. Additionally, previous work has also touched on these two problems. Based on these points, I think the summary of the first contribution is inappropriate.

**Questions:**

In addition to the aforementioned weak points, I have the following question:

1)	Besides Degree of Consistency, are there other evaluation criteria for the model's explainability? Can it be demonstrated in a more intuitive way?

**Reviewer Confidence:**

3: The reviewer is confident but not certain that the evaluation is correct

**Scope:**

4: The work is relevant to the Web and to the track, and is of broad interest to the community

---

### Official Review · Reviewer_TdTr · 2023-11-23

**Novelty:** 5
**Technical Quality:** 5

**Review:**

Summary:

This paper explores non-identifiability and explainability overfitting issues in cognitive diagnosis (CD)-based learner modelling and proposes an Identifiable Cognitive Diagnosis Framework (ID-CDF) to address these challenges. The Identifiable Cognitive Diagnosis Model (ID-CDM) is introduced as an implementation of ID-CDF, demonstrating promising results on benchmark datasets.

Pros:

1. The paper presents a well-researched and comprehensive approach to personalized learner modelling.
2. It is generally well-written and easy to follow.

Cons:

In the EXPERIMENT section, while ID-CDM excels in IDS and DOC indicators (measuring identifiability and explainability), it lacks an evaluation of recommendation performance. Although the emphasis of this paper is on identifiability and explainability in personalized learner modelling, the ultimate goal is the accurate recommendation. It remains unclear if prioritizing identifiability and explainability in personalized learner modelling directly translates to improved recommendation accuracy.

**Questions:**

Can prioritizing Identifiability and explainability in personalized learner modelling lead to enhanced recommendation performance?

**Ethics Review Description:**

No ethical issue.

**Reviewer Confidence:**

2: The reviewer is willing to defend the evaluation, but it is likely that the reviewer did not understand parts of the paper

**Scope:**

3: The work is somewhat relevant to the Web and to the track, and is of narrow interest to a sub-community

---

### Official Review · Reviewer_Axw1 · 2023-11-24

**Novelty:** 5
**Technical Quality:** 5

**Review:**

Pros
1.	The paper studies the fundamental yet significant task in personalized learner modeling.
2.	They perform visualization on the clustering of learner traits diagnosed by different models.
Cons
1.	The technical contributions are minor when the response-proficiency-response paradigm is inspired by the encoder-decoder models.

**Questions:**

1.	Could you please explain more motivation behind the response-proficiency-response paradigm?

**Reviewer Confidence:**

4: The reviewer is certain that the evaluation is correct and very familiar with the relevant literature

**Scope:**

4: The work is relevant to the Web and to the track, and is of broad interest to the community

---

### Official Review · Reviewer_Eovw · 2023-11-30

**Novelty:** 5
**Technical Quality:** 5

**Review:**

Personalized learner modeling using cognitive diagnosis (CD), which aims to model learners' cognitive states by diagnosing learner traits from behavioral data, is a fundamental yet significant task in many web learning services. Existing methods that follow the proficiency-response paradigm face two issues: non-identifiability and explainability overfitting. This paper investigates these problems and proposes a new response-proficiency-response (R-P-R) paradigm to address the two problems from their roots. Based on this paradigm, an ID-CDF (Identification Cognitive Diagnosis Framework) is constructed. Experimental results also demonstrate the advantages of this approach.

Pros:
1. The paper is well organized and has a clear presentation with exquisite schematic diagrams.
2. The proposed method in this draft is easily understandable and exhibits exceptional practicality.
3. The code for this paper is open-source, making it easy to analyze and reproduce.

Cons:
see as the questions

**Questions:**

1. How does the proposed method perform in terms of efficiency? The authors may consider including a comparison of the runtime of each method to demonstrate the efficiency of the proposed approach.

**Reviewer Confidence:**

3: The reviewer is confident but not certain that the evaluation is correct

**Scope:**

4: The work is relevant to the Web and to the track, and is of broad interest to the community

---

### Decision · Program_Chairs · 2024-01-22

**Decision:**

Accept

**Comment:**

The paper proposes to improve on existing cognitive diagnosis frameworks for personalized learner modeling by specifically aiming to improve identifiability and what the authors refer to as explanation-overfitting.

 Reviewer comments span a large range of questions and concerns and I think the authors provided an informative rebuttal.

 As the current reviewer position stands, the paper is borderline, so I had a read through it myself to see if I could tip it to accept. I provide the following comments.

 Sharing a concern with one reviewer who asks if there are other measures of explainability, I find that the authors have a very indirect measure of explainability that appears to me to be more of a performance measurement. In the rebuttal, they explain / defend their DOC explainability measurement as follows:

 > if learners' diagnostic results are explainable, then learners with higher/lower diagnostic knowledge proficiencies should have higher/lower response scores for each question.

 My key concern here is that explainability appears to be measured in aggregate over all knowledge concepts c_k, so it's hard to get a sense of the individual explainability of a student's diagnostic knowledge proficiency for each c_k (which is ultimately what I think a human would want to look at). Wouldn't it make more sense to look at all questions Q_{c_k} requiring knowledge component c_k and provide a scatterplot (one point for each student) where you plot the student's performance on Q_{c_k} vs. their learned proficiency score for c_k? An *explainable* proficiency score for c_k should show reasonable correlation with student performance on those questions. Moreover, for purposes of human interpretability, I would argue there would ideally be a linear relationship in this scatterplot rather than just a monotonic relationship between pairs of students as measured in DOC. As is, the current numeric explainability metric DOC only seems to indirectly assess what would be directly viewed and interpreted by humans. I get how DOC (and a monotonicity test) is a step in the right direction for explainability, but to me it falls a little short of the mark for a paper that puts "explainability" in the title and makes it a core objective of the paper.

 In a separate vein, I also share concerns with reviewers where they request additional justification, motivation, and discussion of content that was omitted from the paper. In response to these questions, the authors have provided three responses like the following: "Due to limited pages, we did not discuss this in detail in the paper." While the Appendix is not required reading material for reviewers, this is where all of this extra information should have been placed and referenced so that the paper and Appendix make for a self-contained paper that can be read standalone. As is, I feel there is substantial critical explanatory information provided in the rebuttal that is not presently in the submitted paper or Appendix.

 Overall, I believe this is a very solid and well-structured paper that makes progress in the field of cognitive diagnosis, but I also believe that it falls a little short in fully measuring human-interpretable explainability and omits a lot of explanatory content that is important for TheWebConf audience.